# Programmable adhesion and morphing of protein hydrogels for underwater robots

Sheng-Chen Huang [1], Ya-Jiao Zhu[1], Xiao-Ying Huang[1], Xiao-Xia Xia [1] ✉ & Zhi-Gang Qian [1] ✉

Soft robots capable of efficiently implementing tasks in fluid-immersed environments hold great promise for diverse applications. However, it remains challenging to achieve robotization that relies on dynamic underwater adhesion and morphing capability. Here we propose the construction of such robots with designer protein materials. Firstly, a resilin-like protein is complexed with polyoxometalate anions to form hydrogels that can rapidly switch between soft adhesive and stiff non-adhesive states in aqueous environments in response to small temperature variation. To realize remote control over dynamic adhesion and morphing, $Fe_3O_4$ nanoparticles are then integrated into the hydrogels to form soft robots with photothermal and magnetic responsiveness. These robots are demonstrated to undertake complex tasks including repairing artificial blood vessel, capturing and delivering multiple cargoes in water under cooperative control of infrared light and magnetic field. These findings pave an avenue for the creation of protein-based underwater robots with on-demand functionalities.

Underwater soft robots hold great promises for diverse applications ranging from miniaturized medical tools[1,2], sensors[3] to ocean exploration[4], due to their intriguing capability to navigate and operate in fluid-immersed environments. Smart materials are increasingly engineered as actuators and sensors to construct soft robots, such as liquid crystal polymers[5], dielectric elastomers[4] and stimuli-responsive hydrogels[6–10]. However, these robots are inevitably faced with a set of challenges that limit their performances and applications[11]. One of the toughest challenges is to achieve controllable and strong adhesion to diverse surfaces in aqueous environments[12]. This adhesive property is essential for the soft robots to efficiently implement various underwater tasks such as remotely collecting bio-samples, transmitting bio-signals, and sticking to tissue damages for healing and repair[13–15].

Underwater adhesion is technically challenging for most soft materials because formation of a hydration layer impedes the interaction between substrates and adhesive materials[16,17]. Inspired by marine organisms which are able to bond dissimilar materials in aquatic environments by secreting adhesive proteins or polypeptides[18–21], various underwater adhesives have been biomimetically designed and developed[22–26]. Over the past decade, the most common method for formulating these underwater adhesives is to incorporate L−3,4-dihydroxyphenylalanine into the molecular design of materials. Nevertheless, such adhesives usually lack dynamically responsive properties and are prone to lose adhesiveness due to the over oxidation of catechol groups[27]. An alternative method for developing adhesives uses the complex coacervation of oppositely charged polyelectrolytes via liquid–liquid phase separation[24,25,28]. Benefiting from dynamic nature of noncovalent interactions, these adhesives are endowed with tunable adhesion in response to environmental stimuli[22]. However, the reported underwater adhesives generally have poor mechanical properties and lack controllable attachment/detachment behavior over diverse substrate surfaces[29,30], which were not able to accomplish complex robotic tasks such as catching, locomotion and shape morphing.

In this study, we develop a gel-type underwater adhesive through controllable complexation between intrinsically disordered resilin-like

[1]State Key Laboratory of Microbial Metabolism, Joint International Research Laboratory of Metabolic and Developmental Sciences, and School of Life Sciences and Biotechnology, Shanghai Jiao Tong University, 800 Dongchuan Road, Shanghai 200240, People's Republic of China. ✉e-mail: xiaoxiaxia@sjtu.edu.cn; zgqian@sjtu.edu.cn

proteins (RLPs)[31–33] and Keggin-type polyoxometalates[34,35]. The resulting adhesive hydrogels can rapidly switch between non-adhesive rigid state and adhesive soft state within a narrow and mild temperature range. This feature enables the development of soft robots functionalized with magnetic $Fe_3O_4$ nanoparticles that can realize switchable adhesion, locomotion and shape morphing in aqueous environments under cooperative control of infrared (IR) light and magnetic field. The underwater robots have also been demonstrated to undertake complex tasks including capturing and delivering multiple cargoes, and repairing artificial blood vessel, which thus show great potential for biomedical applications.

## Results and discussion
### Design principles to engineer dynamic adhesive protein hydrogels
RLPs are an important type of intrinsically disordered proteins which are positively charged, and chemically crosslinkable due to the presence of characteristic tyrosine residues[36,37]. We postulated that these proteins could electrostatically interact with silicotungstic acid (SiW), a typical Keggin-type polyanion to form complex coacervates for the construction of a dynamic underwater adhesive. The inorganic nanocluster SiW attracts our attention because it has a rigid framework, well-defined topology and high ionization propensity in water to form quadrivalent electrostatic interactions with positively charged polymers[38]. To test this hypothesis, R32 protein composed of 32 repeats of the resilin-like block (GGRPSDSYGAPGGGN) is chosen as a model, which contains a number of crosslinkable (tyrosine) and charged (arginine) amino acids (Fig. 1a). This protein was recombinantly produced and purified as described in the "Methods". SDS-PAGE analysis revealed that the purified protein had at least 95% purity (Supplementary Fig. 1a), and mass spectrometry further verified its identity (Supplementary Fig. 1b). For the complexation reaction, it is preferred to pre-crosslink R32 protein via di-tyrosine sites to increase the cohesion of the resulting complexed coacervates.

The complexation process was performed in two key steps (Fig. 1b). The first step was to drop the pre-crosslinked R32 protein solution into a SiW bath in which the coacervate droplets were formed due to the fast complexation reaction between R32 protein and SiW. The second step was gentle stirring which would fuse the separated droplets into aligned fibrils for the formation of a sticky and soft hydrogel. To prove that SiW was incorporated into the resulting hydrogels by the complexation reaction, Fourier transform infrared (FTIR) spectroscopy and [183]W nuclear magnetic resonance (NMR) spectroscopy of the hydrogels were performed. The FTIR spectra showed that characteristic vibration bands of SiW existed in our hydrogels and slightly shifted in comparison with those of the parent SiW (Supplementary Fig. 2a), demonstrating the formation of SiW-involved electrostatic interactions in the complexed hydrogels[26]. The structure of SiW within the hydrogels was also demonstrated to be well-retained by the [183]W NMR spectra analysis (Supplementary Fig. 2b).

The as-obtained protein hydrogels displayed an intriguing shape moldable property at room temperature, which could be remolded into various arbitrary shapes including a sphere, cube, rope and leaf (Fig. 1c). This feature was conducive to the seamless contact of the hydrogel to other substrates even with rough surfaces, thereby enhancing the adhesion capability. More interestingly, the complexed protein hydrogel exhibited aligned microfibrillar structures as revealed by scanning electron microscopy (Fig. 1d), which resemble many natural underwater adhesives[39]. These hierarchically assembled fibrillar structures are usually harnessed to achieve strong and robust interfacial adhesion under dynamic and turbulent environments[40].

### Thermo-switchable mechanical properties of the hydrogels
Having established the complexation method to fabricate R32 hydrogels, we then explored whether the dynamic electrostatic interactions could impart thermoresponsive properties to the hydrogels. Firstly, mechanical properties of the protein hydrogels were measured over a broad temperature range. Considering that pre-crosslinking degree of R32 might affect the resultant hydrogel's mechanical properties, four hydrogels corresponding to 0, 5%, 15% and 30% di-tyrosine crosslinks of R32 were studied, which were denoted as R32-SiW, R32-5%-SiW, R32-15%-SiW and R32-30%-SiW, respectively (Supplementary Fig. 3). These hydrogels had comparable water content of ~20%, and variable SiW-to-R32 ratio that depended on the prior di-tyrosine crosslinking degree of the R32 protein (Supplementary Table 1). With an increase in the chemical crosslinking degree, the SiW-to-R32 ratio decreased from ~6.4:1 to 4.9:1, indicating compositional tunability of the double crosslinked hydrogels. Furthermore, we studied stability and degradation of these hydrogels in water (Supplementary Fig. 4). Interestingly, all the gels were very stable in water over an extended time period of 240 h at the ambient and lower temperatures (10–25 °C). At the body and higher temperatures, these hydrogels well retained their weights within the first 10 h, yet extended soaking partially eroded the hydrogels to extents that depended on the di-tyrosine crosslinking degrees. Overall, these results demonstrated another level of tunability (underwater stability) of the protein hydrogels by prior chemical crosslinking.

According to dynamic mechanical analysis (DMA), the storage modulus ($E'$) of the R32 hydrogels were temperature dependent, and moderately decreased with increasing temperature from 0 to 80 °C (Fig. 2a). This change should be attributed to the introduced electrostatic interactions because the single di-tyrosine crosslinked R32 hydrogel maintained constant modulus across the entire temperature range (Supplementary Fig. 5a). In addition, the change in modulus upon heating is highly reversible as revealed by the cyclic DMA measurement (Supplementary Fig. 5b), indicating the potential controllability on gel mechanics by temperature. The softening temperatures ($T_s$) of the hydrogels, which were defined as the peak temperatures of the loss factor (tan δ) curves, were found to be in the room temperature range from ~20.4 to 26.7 °C (Fig. 2a). In addition, there is a positive relationship between the $T_s$ and the degree of di-tyrosine crosslinking in the hydrogels (Supplementary Table 1).

The thermo-softening of the hydrogels was also observed in uniaxial tensile tests (Fig. 2b). Below the $T_s$, the hydrogels were rigid with high Young's modulus and low breaking strain levels, whereas above the $T_s$, they were soft with low Young's modulus and high breaking strain values. To our knowledge, such a sharp change in the mechanical stiffness under a mild room-temperature range has rarely been reported for the hydrogels, which might be due to the peculiar electrostatic interactions between the positively charged R32 and polyanion SiW. When heated to above $T_s$, the equilibrium of electrostatic complexation shifted toward the disassociated state, thus lowering the physical crosslinks of the network and reducing the mechanical stiffness. Upon cooling to below $T_s$, the equilibrium shifted toward the associated state, thus reforming the densely crosslinked network and enhancing the mechanical stiffness. This controllable thermodynamic equilibrium allowed the gels to exhibit two distinct mechanical states in wet environments: rigid state (below $T_s$) and soft state (above $T_s$) (Fig. 2c). It is noted that this dynamic ionic bond can also impart the complexed hydrogel with thermal healing properties, and the healing efficiency increases markedly with healing times and temperature (Supplementary Fig. 6).

The photographs in Fig. 2d and Supplementary Movie 1 vividly illustrated the thermo-switchable mechanical properties of the complexed hydrogel. At 20 °C, a knife-shaped hydrogel (~6.5 g) with a thickness of 2 mm could penetrate a fresh crisp apple (~100 g), and then steadily withstand the apple in the air in a single cantilever mode (i–iii). During the whole process, no obvious deformation of the hydrogel was observed, which implied that the gel material was rigid and strong with good load-bearing capacity at this temperature. When

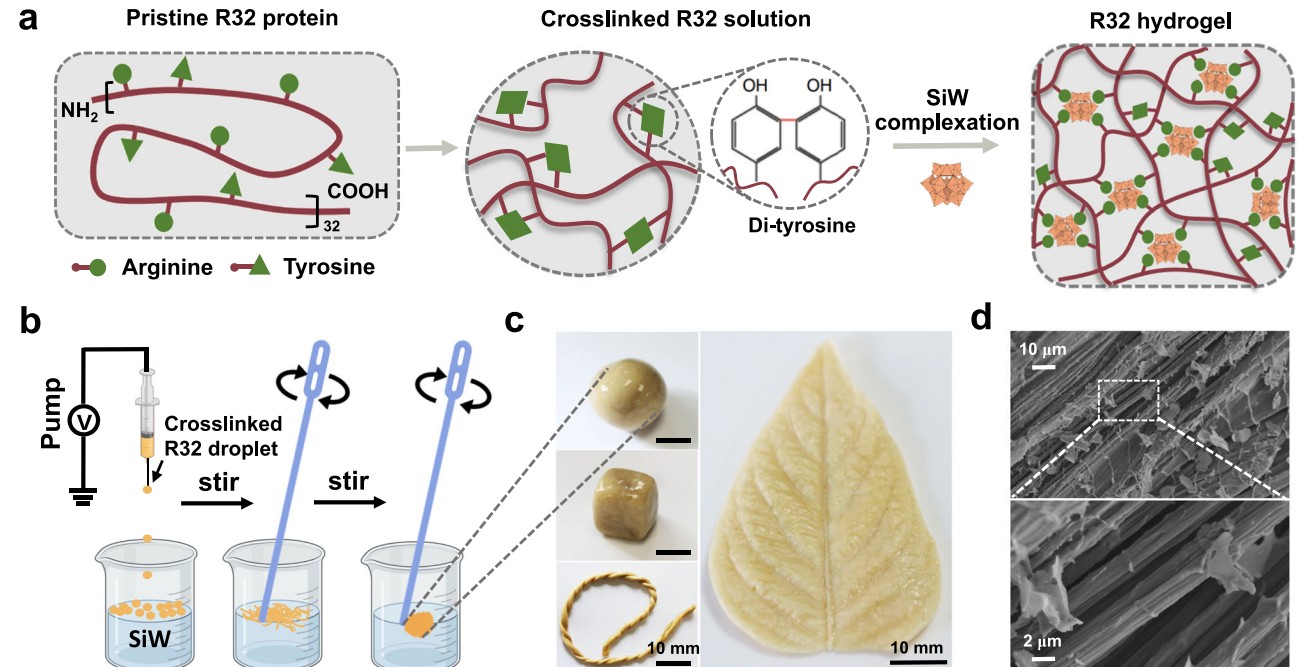

**Fig. 1 | Design and fabrication of silicotungstic acid (SiW)-complexed R32 protein hydrogels. a** Proposed design principle of the R32 hydrogels. **b** Schematic illustration of the key process for complexation between crosslinked R32 and SiW. Created with BioRender.com. **c** The as-prepared ball-like hydrogel can be remolded into different shapes, such as a cube, a rope and a leaf. **d** SEM images of the microfibril structures in the hydrogels. Data in (**c**) and (**d**) are representative of $n = 2$ independent experiments.

heated to 50 °C, the knife-shaped hydrogel instantly became soft and could not support the apple's weight. With the apple falling down to the ground, the hydrogel was stretched to at least three times its original length (iv). When the temperature was decreased from 50 °C to 20 °C again, the soft and viscoelastic hydrogel immediately converted back into its strong and rigid state. Upon removal of the hand, the gel could self-support its weight and maintain the deformed shape in the air (v).

**Thermo-switchable underwater adhesiveness**

We next explored whether the complexed hydrogels exhibit underwater adhesive property. As a typical example, the R32-5%-SiW hydrogel was tested for adhesion to diverse substrate surfaces in a water bath. The results showed that the gel could in situ adhere to both hydrophilic and hydrophobic surfaces, and even biological tissues at 25 °C. The adhesive strengths measured by probe-tack tests reached 30–40 kPa (Supplementary Fig. 7), which are comparable to most of the reported bioadhesive hydrogels[41].

To further investigate whether the dynamic electrostatic interactions could impart thermoresponsive adhesiveness to the hydrogels, the adhesive strengths of the hydrogels were measured over a broad temperature range (Fig. 3a). Interestingly, all the hydrogels exhibited a similar trend in the evolution of the adhesion strength with increasing temperature, which rose to a maximum and then decreased to a low value. We assumed this change was attributed to the trade-off between adhesion and cohesion of the hydrogels. It appears that when the adhesion and cohesion of the hydrogel reached a balance, the gel could acquire the strongest adhesiveness. In addition, the temperature at which the hydrogel reached the maximum adhesiveness could be finely controlled by tuning the degree of di-tyrosine crosslinking in the hydrogels.

Furthermore, the temperature at which the hydrogel initiated adhesion ($T_i$) was determined using a water bath at a temperature interval of 0.5 °C (Supplementary Fig. 8). In general, the hydrogel $T_i$ increased markedly with an increase in the degree of di-tyrosine

crosslinking. For example, the R32-SiW and R32-5%-SiW hydrogels possessed $T_i$ values at -23.7 °C (Supplementary Table 1), very close to their $T_s$ values, which indicated that these hydrogels exhibited two distinct states of soft adhesive and stiff non-adhesive. In contrast, the $T_i$ values of the R32-15%-SiW and R32-30%-SiW hydrogels reached 43.5 °C and 52.7 °C, respectively, much higher than their respective $T_s$ values, which was indicative of one additional hydrogel state (soft but non-adhesive) between $T_s$ and $T_i$. This might be due to that higher di-tyrosine crosslinking in the hydrogels caused less availability and exposure of the adhesive groups for participation in adhesion to the substrate surface. As vividly displayed in Fig. 3b, the R32-5%-SiW hydrogel as an example showed a thermo-switchable adhesiveness. The gel could not adhere to the surface of the steel weight underwater at 20 °C, but could adhere and lift the weight out of the water at 37 °C. With the water bath temperature increased to 70 °C, the adhesion could also occur, but the lifting of the steel weight failed because of cohesion collapse of the gel at this fairly high temperature.

On the basis of the above results, we have developed a temperature switch method to increase underwater adhesion of the hydrogels. This method includes two steps that are warm attaching at a temperature higher than $T_i$, followed by cooling to a temperature lower than $T_i$ (Fig. 3c). To demonstrate this strategy, the evolutions of the underwater adhesion strengths of the hydrogels were recorded upon cooling to different temperatures after warm attachment at 25 °C, 45 °C and 70 °C, respectively. Surprisingly, all the hydrogels could be endowed with an order of magnitude increase in underwater adhesiveness at 4 °C via the warm attachment at a certain temperature higher than their respective $T_i$. In general, higher attachment temperature ($T_A$) and lower cooling temperature for each hydrogel would result in higher adhesiveness (Fig. 3c). Notably, the hydrogel with higher degree of di-tyrosine crosslinking exhibited significantly higher adhesiveness at room temperature although higher $T_A$ was required to initiate the adhesion (Fig. 3d). As vividly displayed in Supplementary Fig. 7d, the R32-30%-SiW hydrogel with an adhesion area of ~3.14 cm²

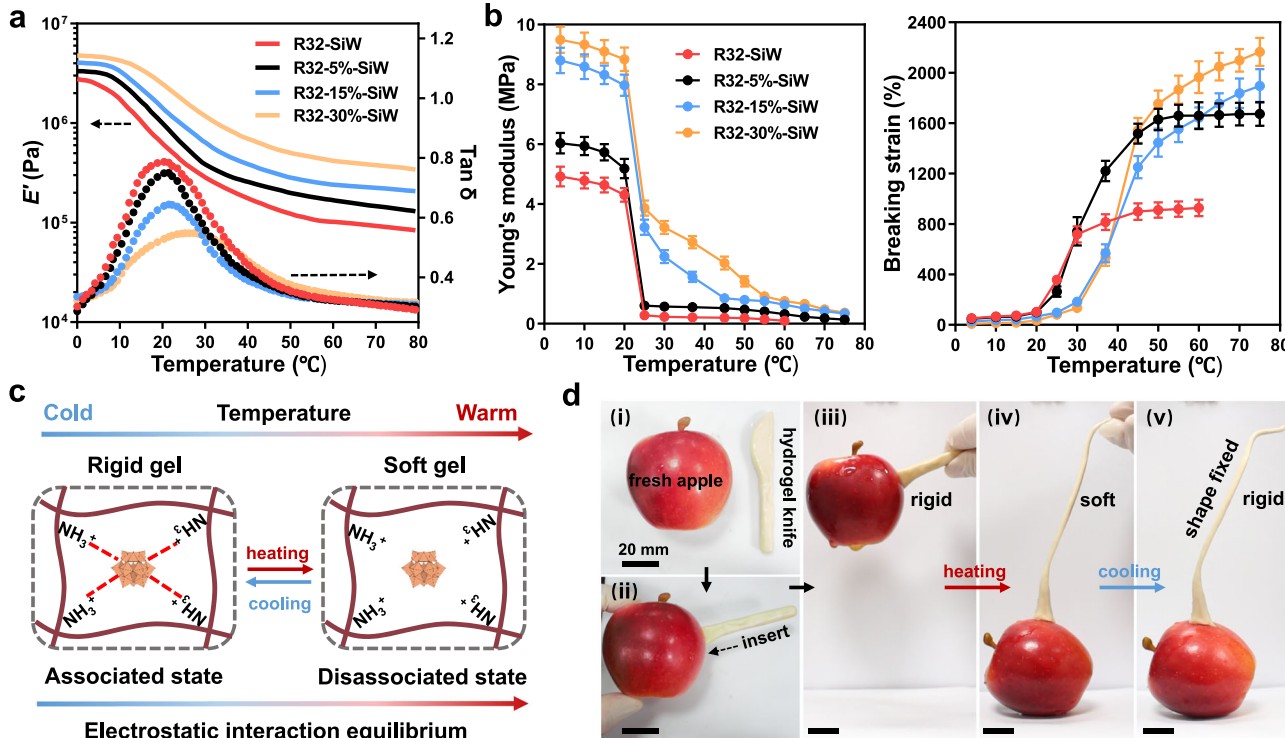

**Fig. 2 | Temperature-dependent mechanical properties of the complexed R32 hydrogels. a** The storage modulus ($E'$) and loss factor (tan δ) as a function of temperature of the hydrogels. R32-SiW, R32-5%-SiW, R32-15%-SiW and R32-30%-SiW correspond to the SiW-complexed hydrogels with a di-tyrosine crosslinking degree at 0, 5%, 15% and 30%, respectively. **b** The Young's modulus and breaking strain of the hydrogels measured by tensile tests at various temperatures. The error bars represent the standard deviations of triplicates. **c** The proposed mechanism of the switchable mechanical properties of the hydrogels. **d** The photographs demonstrating that the hydrogel (5% di-tyrosine crosslinking degree) can repeatedly switch between rigid (20 °C) and soft state (50 °C). Scale bar, 20 mm. Data in (**a**) and (**d**) are representative of $n = 2$ independent experiments. Data in (**b**) are presented as mean ± s.d. of $n = 3$ independent samples, and are representative of two independent experiments. Source data are provided as a Source Data file.

could easily hang up a weight of 15 kg underwater at 20 °C via warm attachment at 70 °C.

Considering both the mechanical properties and underwater adhesion, we found that the R32-5%-SiW hydrogel could switch between soft adhesive and stiff non-adhesive states. Indeed, this gel reversibly switched between the stiff and soft states within 9 s in response to a temperature variation of 10 °C (Supplementary Fig. 9a). Such a rapid switch in mechanical properties might be attributed to the fast dissociation-reassociation of dynamic electrostatic interactions between SiW and the positively charged residues of R32 protein. We also found that the soft gel network was more dynamic with higher chain mobility than its rigid counterpart (Supplementary Fig. 9b). In addition, the surface of the underwater soft gel at 30 °C was fairly hydrophobic, with an oil contact angle of only 43° (Supplementary Fig. 9c), whereas the gel surface was relatively hydrophilic at 20 °C.

Based on the above results, we propose mechanisms that underlie the thermally switchable adhesion behavior of the R32-5%-SiW hydrogel (Supplementary Fig. 9d). It is reasoned that the low chain mobility of the hydrogel at the stiff state hampers its close contact to the substrate surface, and the hydrophilic surface of the stiff gel would trap a significant amount of water leading to shielding of protein adhesive groups to closely interact with the substrate surface. In contrast, the thermally-triggered switch of the gel into the soft state with high chain mobility and hydrophobic surface would benefit initiation of strong interfacial adhesion to the substrate of interest. Collectively, the significant variations in chain mobility and surface hydrophobicity contribute to the switchable adhesion behavior.

## Photothermally controlled gel mechanics and underwater adhesiveness

Due to the low energy conversion efficiency of water, it is difficult to locally regulate the temperature of the adhesive hydrogels in aquatic environment[42]. To enable a remote control of the mechanics and underwater adhesion, the hydrogels were incorporated with photothermally-responsive magnetic $Fe_3O_4$ nanoparticles (M-$Fe_3O_4$ NPs). Upon IR illumination, the M-$Fe_3O_4$ NPs can generate strong plasma and convert light into heat, resulting in a rapid increase in the local temperature of the hydrogels. Thereby, we envision that the adhesive hydrogels incorporated with M-$Fe_3O_4$ NPs could be remotely modulated via the control of IR light. In our experiment, R32-5%-SiW hydrogel was selected to fabricate the photothermally controlled adhesive hydrogel because of its consistent and mild $T_s$ as well as $T_i$ (-20 °C). In addition, the integrated M-$Fe_3O_4$ NPs did not influence the mechanical properties or underwater adhesiveness of the gel. The resultant hydrogel was denoted as M-R32-5%-SiW, and its photothermal performance in water was quantified upon IR light illumination (Supplementary Fig. 10). The temperature of the underwater gel could be elevated by 12 to 33 °C within 3 min, reaching temperature values that well surpassed its $T_s$ and $T_i$. This feature is of particular interest as it would allow the gel to morph and adhere in a remotely controllable manner. Figure 4a illustrated the key procedure of remotely regulating the adhesiveness of M-R32-5%-SiW hydrogel. Firstly, the hydrogel was placed on a flat substrate in a water bath at 20 °C. Under exposure to IR light, the hydrogel was heated above 20 °C with a transformation from a rigid non-adhesive state to a soft adhesive state, and adhered to the substrate surfaces. After switching off the light, the gel converted back to the rigid state as soon as the

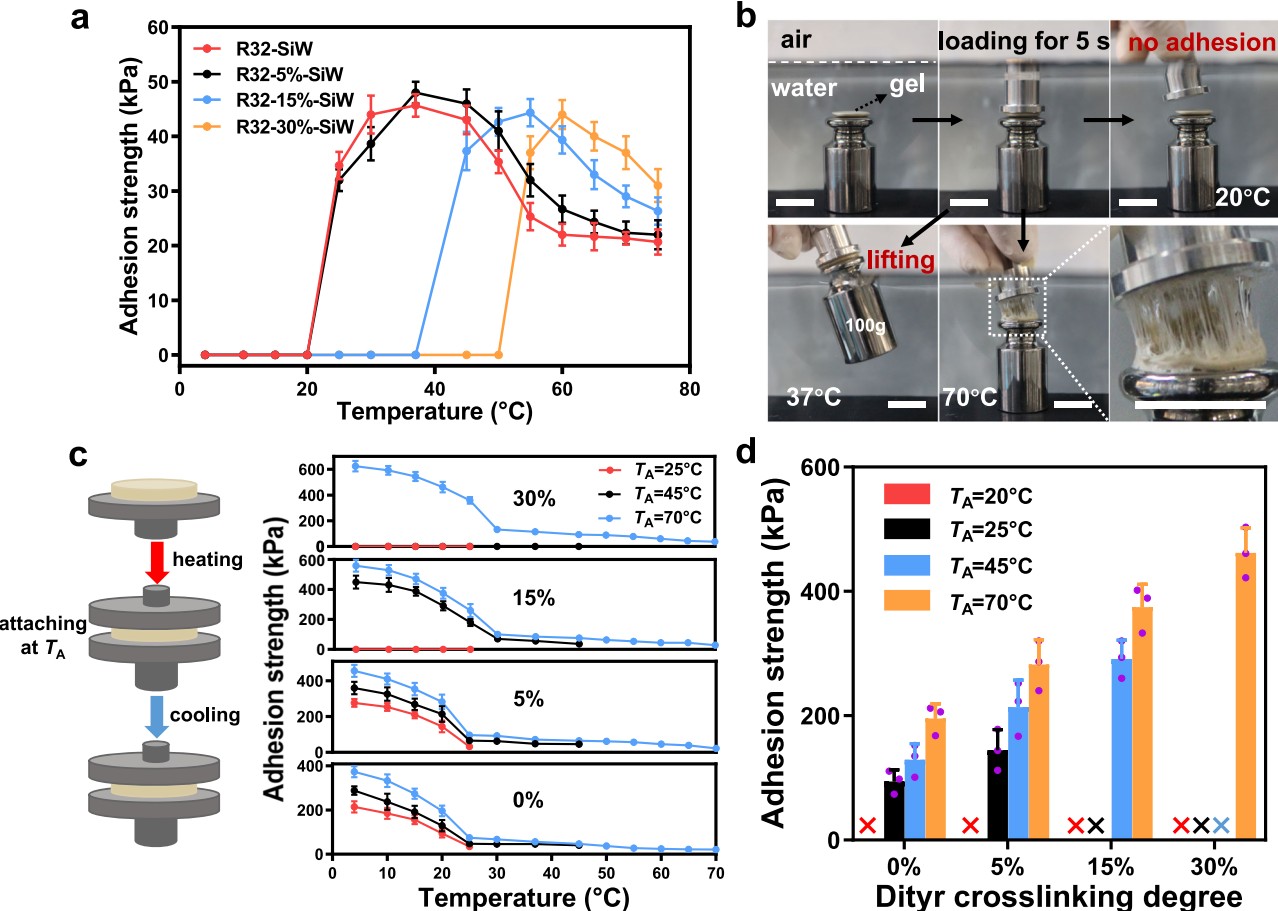

**Fig. 3 | Dynamic underwater adhesion of the SiW complexed R32 hydrogels.**
**a** Underwater adhesion strength evolution of the hydrogels versus temperature measured by probe-tack tests using stainless steel as a model substrate. **b** The adhesion behaviors of R32-5%-SiW hydrogel at 20 °C, 37 °C and 70 °C, respectively. Scale bar, 20 mm. **c** Schematic illustration of the warm attaching and cooling procedure for achieving enhanced adhesion (left). The hydrogels were first loaded onto the steel surface at the indicated attaching temperature ($T_A$) for 30 s and then tested for underwater adhesion at temperatures ranging from 4 to 70 °C (right). **d** Comparison of the underwater adhesion strength of the hydrogels measured at 20 °C after attaching for 30 s at the respective $T_A$ as indicated in (**d**). The cross signs represent that the adhesion strength values were undetectable. Data in (**a**), (**c**) and (**d**) are presented as mean ± s.d. of $n$ = 3 independent samples. Data in (**a**–**d**) are representative of two independent experiments. Source data are provided as a Source Data file.

temperature decreased to 20 °C, and exhibited a higher adhesion with the surfaces.

To demonstrate the feasibility of this controlling process, we monitored the evolutions of the temperature, Young's modulus and in situ underwater adhesion of the gel during switching on and off the IR light (Fig. 4b). It can be found that the local temperature of the gel rose from ~20 to 45 °C within ~180 s under light irradiation and then decreased to ~20 °C within another ~270 s after turning off the light. During this process, the Young's modulus of the gel first decreased from ~6.8 to 0.5 MPa and then restored the original gel stiffness, while the underwater adhesion strength of the gel first increased from 0 to 50 kPa and then decreased back. In addition, there is no noticeable decay in evolution of the Young's modulus and underwater adhesion of the gel during multiple cycles of measurement. The above results confirmed that the photothermal responsiveness of the gel was fully reversible. Figure 4c displayed the IR light-triggered softening of the M-R32-5%-SiW hydrogel in water. As can be observed, the rigid thin gel bar (2 mm × 2 mm × 30 mm, 2.3 g) can easily withstand a ring (10 g) in the single cantilever mode without obvious deformation or breaking within 2 h in a 20 °C water bath. Upon IR light illumination for 15 s, the gel became soft and gradually bent downward. Figure 4d and Supplementary Movie 2 demonstrated the IR light-triggered dynamic underwater adhesion of the M-R32-5%-SiW hydrogel. Initially, a piece of the gel (2 mm in height and 0.2 g in weight) anchored on a spherical

object (40 mm in diameter and 200 g in weight) was submerged in the water bath at 20 °C. Without the IR light illumination, the gel could not adhere to the stainless steel probe. Upon the illumination for 15 s, the probe could rapidly adhere to the gel, and then lift the object out of the water bath after turning off the light.

## Development of biomimetic underwater adhesive robots
Among predators using an adhesive tongue to feed, chameleons are able to capture insect prey located up to 1.5 body lengths away in air by projecting the tongue which elongates more than six times its resting length[43–45]. Once the viscous tongue tip is in contact with a prey, a strong adhesion is formed at the interface to ensure a successful capture (Fig. 5a). Inspired by such a biosystem, the M-R32-5%-SiW hydrogel was engineered as an artificial chameleon's tongue to mimic the preying process. It should be noted that our designed robotic tongue could hunt underwater which is beyond nature.

Having demonstrated that the stiffness and underwater adhesive behavior of the hydrogel can be modulated by IR light irradiation. To further realize elongation deformation of the robotic tongue, we introduce magnetic actuation to effectively and remotely control the shape-morphing of the gel when it is in soft state. Figure 5b illustrates the proposed concept of a chameleon tongue inspired (CTI) robot capturing a prey from a distance in water under cooperative application of IR light illumination and magnetic field. The CTI robot is first

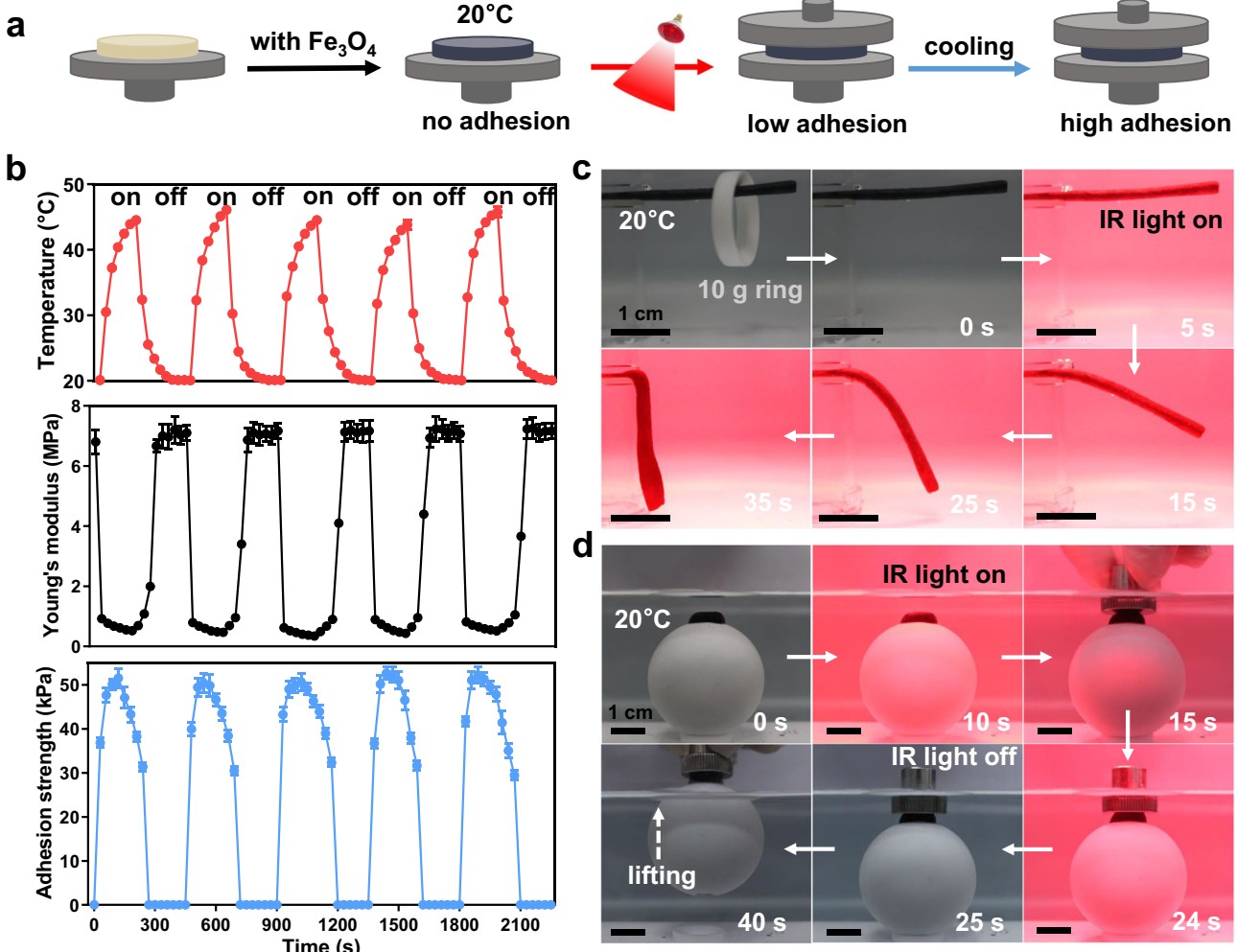

**Fig. 4 | Photothermally-responsive mechanical properties and underwater adhesion of the magnetic nanoparticles incorporated R32-5%-SiW hydrogel (M-R32-5%-SiW). a** Schematic diagram showing infrared (IR) light-responsive switching of the hydrogel adhesion. **b** The evolution of the temperature, Young's modulus and underwater adhesion of the hydrogel during switching on and off the IR light irradiation. **c** Photographs of the softening of the hydrogel in water upon IR light irradiation. **d** The snapshots of the hydrogel successfully adhering to the stainless steel probe followed by lifting the ball object (200 g) via switching IR light. Data in (**b**) are presented as mean ± s.d. of $n$ = 3 independent samples. Data in (**b–d**) are representative of two independent experiments. Source data are provided as a Source Data file.

anchored on an object A which mimic chameleon's body, and then placed in a liquid environment at 20 °C. The proposed underwater preying process can be divided into three stages. In the first stage, upon IR light illumination, photothermal heating of the magnetic particles causes an increase in temperature of the robotic tongue above its $T_s$, thus softening the robot and initiating its adhesiveness. In the second stage, an external static magnetic field parallel to the direction from the robotic tongue to a prey (object B) is applied coupled with IR heating, the decreasing cohesion is overcome by magnetic field-induced pulling force, driving the elongation of the robotic tongue toward the target object. When the tongue tip reaches the surface of object B, a sticky adhesion is formed at the interface. In the third stage, when the magnetic field and the IR light are removed, the tongue body converts back to rigid state and the extended shape is locked as soon as the temperature decreases to 20 °C, thus leading to a strong adhesion between the two objects.

Figure 5c and Supplementary Movie 3 showed the experimental demonstration of the robotic tongue (5 mm in height and 0.5 g in weight) mimicking capture of a prey underwater (8 mm in in diameter and 1.6 g in weight) from 15 mm away. Upon IR irradiation for 30 s, the robotic tongue was photothermally heated above its $T_s$, which induced

a significant decrease in elastic modulus of the tongue body. Once applying the magnetic field and IR light simultaneously, the softened tongue gradually grew up and elongated toward to object B along the magnetic field direction. After about 500 s, the extended robotic tongue adhered to the object B. The IR light and magnet were then removed to lock the shape of robotic tongue and strengthen the adhesion between the two objects by cooling down. After another 220 s, the object B could be picked up through lifting object A, indicating a successful remote capture.

Next, we quantitatively analyzed the relationship between the elongation strain of the tongue and the final underwater adhesion force between the two objects. The results in Fig. 5d revealed that the adhesion force deceased slowly at the initial stage and then declined sharply after the tongue extended more than four times its resting length. This can be explained by the decreasing cross-sectional area and inhomogeneous deformation of the overstretched gel, which resulted in the decreased cohesion per unit area and breakage of the gel (Fig. 5e). In addition, the extended robotic tongue could be bent toward any desired directions by controlling the direction of the magnetic field. This feature made it possible for the tongue to capture the object located at any place in 3D space underwater (Fig. 5f).

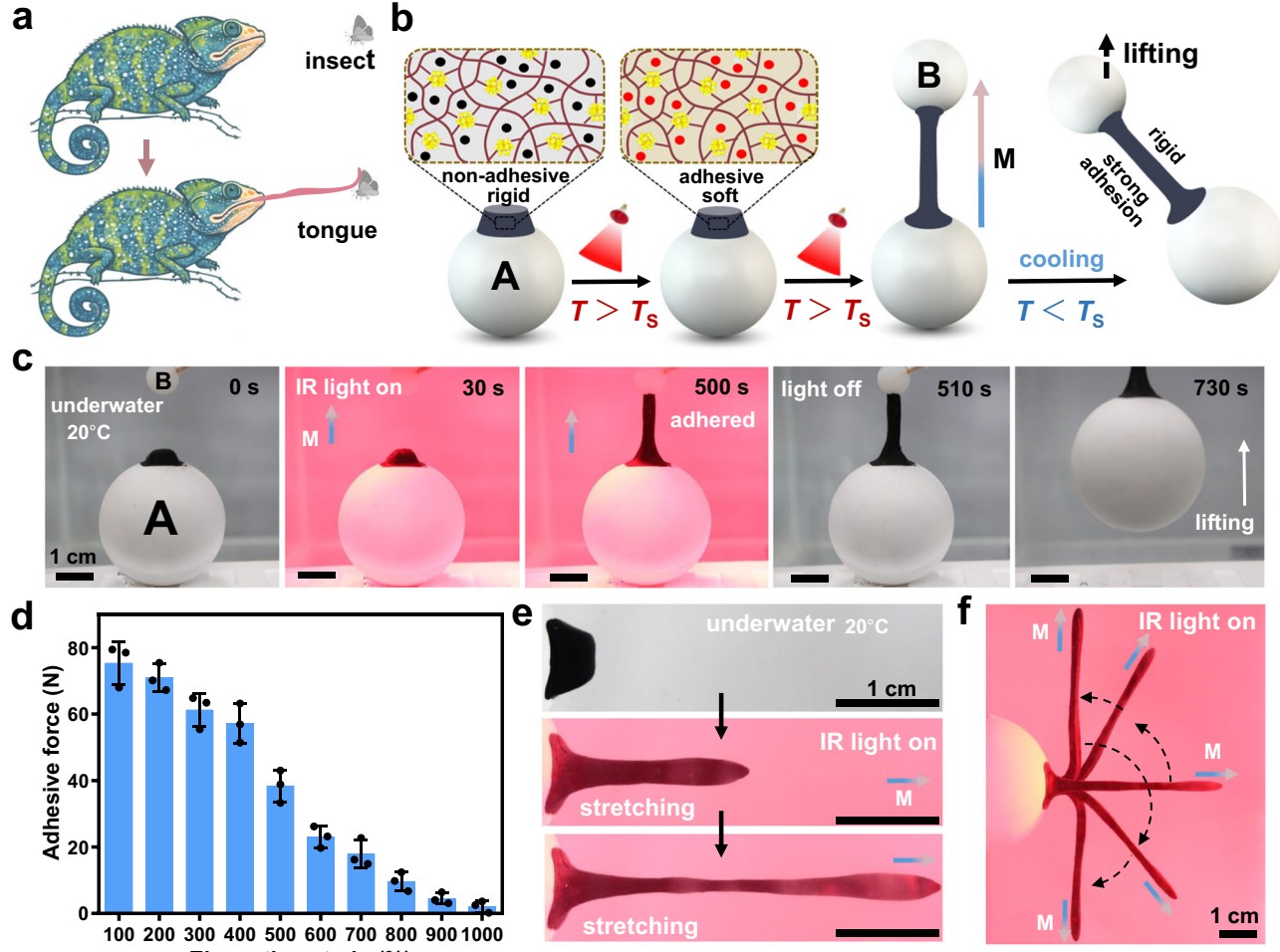

**Fig. 5 | Design and characterization of the chameleon tongue inspired (CTI) robot. a** Schematic illustration of a chameleon hunting an insect through elongating its adhesive tongue. Created with BioRender.com. **b** Schematic process of the CTI robot capturing a prey from a distance in water. Illumination with an infrared (IR) light increases temperature ($T$) of the hydrogel above its softening temperature ($T_s$), and subsequent application of a magnetic field (M) allows morphing of the hydrogel to adhere to an object for lifting operation. **c** Experimental demonstration of the robotic tongue mimicking capture of a prey from a distance underwater. **d** The underwater adhesion force between two objects versus the elongation strain of the robotic tongue. **e** Photographs showing the elongation process of the robotic tongue, and demonstrating that inhomogeneous deformation occurred in the middle section of the tongue when it was over-stretched. **f** Upon IR light irradiation, the extended tongue first bent 90° counter-clockwise and then 180° clockwise with the change in the direction of the magnetic field. Data in (**d**) are presented as mean ± s.d. of $n = 3$ independent samples, with individual data points shown as black dots. Data in (**c**–**e**) are representative of two independent experiments. Source data are provided as a Source Data file.

## Performances of underwater adhesive robots

Considering the exquisite combination of shape-morphing and dynamic adhesion, our magnetic CTI robot was expected to be able to implement complicated tasks in aqueous environments. To demonstrate this, a three-arm robot was fabricated to test its capability of capturing and delivering multiple cargoes in water (Fig. 6a). The detailed experimental procedures were displayed in Fig. 6b. Firstly, the robot and cargoes were placed in water at 20 °C ($t = 0$ s), and then the robot was moved toward the cargoes with a permanent magnet, which was put under the substrate and moved in the same way. Once the robot reached the target position ($t = 15$ s), a vertical IR light illumination on the area between cargo 1 and the tongue nearby was applied. With the cooperative application of a magnetic field, the irradiated tongue gradually elongated and finally captured the cargo 1. Upon removing the light and magnetic field, a strong adhesion was formed between the tongue tip and the cargo. Repeating the above manipulation process, the robot successively captured the cargo 2 and 3. At last, the robot successfully transported the cargoes back to the home position with the magnet.

In our daily life, bleeding is a common occurrence due to the blood vessel damage, and sutures have long been the major choice for hemostasis. However, suturing a blood vessel deep inside the body such as the artery may cause tissue damage. To avoid or reduce the secondary damage, the small-scale adhesive robot that can be remotely controlled may be the potential alternative to repair blood vessel leakage (Fig. 6c). To demonstrate this, an adhesive robot was fabricated to carry out the repairing task in the artificial blood vessel, which was filled with pig blood that had been pre-incubated at a physiological temperature. In order to visualize the actuation of the robot, a transparent poly(methyl methacrylate) (PMMA) tube was used as the artificial blood vessel. For mimicking the bleeding of the blood vessel, a 25-mm-long incision was produced at the bottom of the tube. As illustrated in Fig. 6d, the robot was able to smoothly navigate through the blood vessel and moved to the leakage under a magnetic field. With the IR light turning on, the robot gradually adhered to the internal surface of the vessel and anchored itself in situ. After that, the robot elongated its body along the vessel wall to cover the damage under the cooperative control of IR light illumination and magnetic field. After

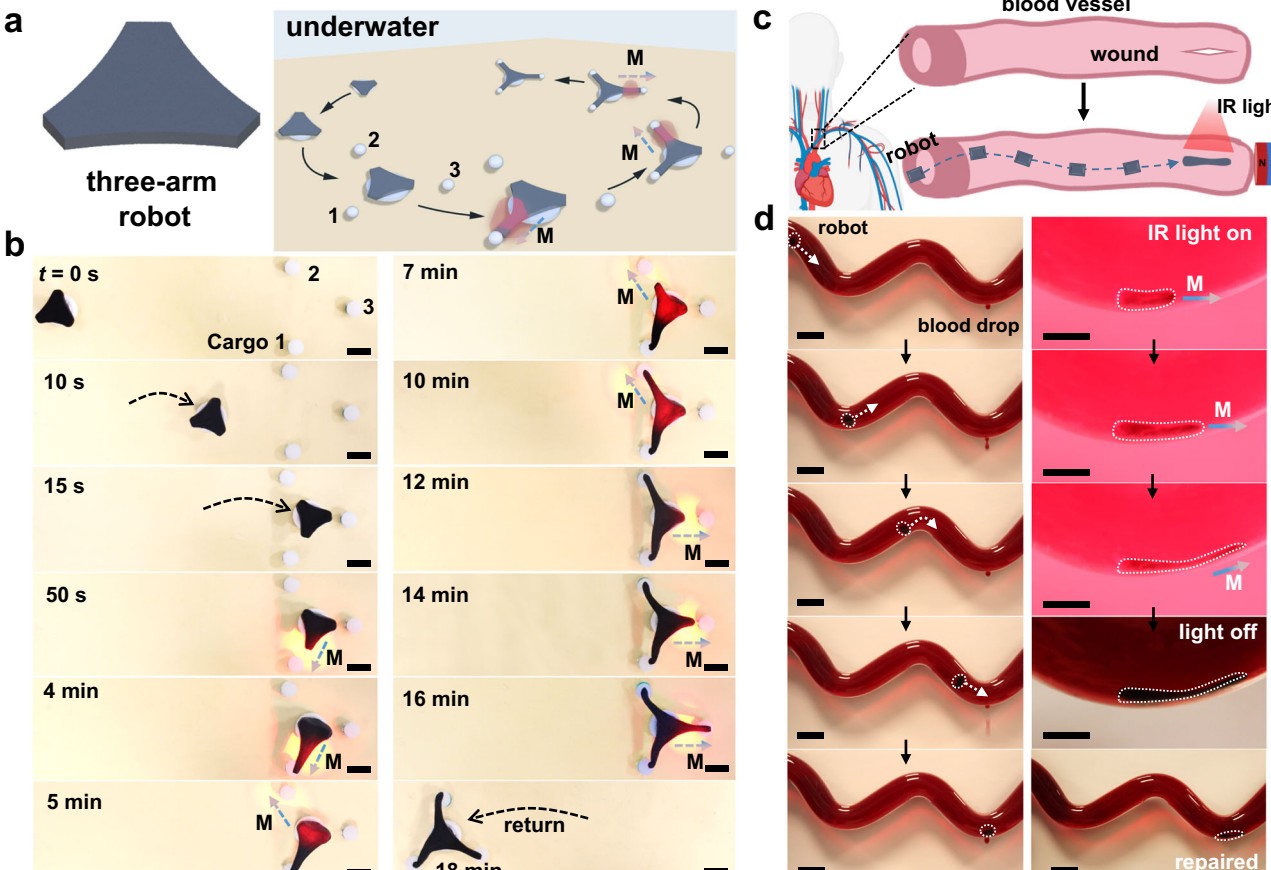

**Fig. 6 | Performances of the underwater adhesive robot fabricated from the M-R32-5%-SiW hydrogel. a** Schematic of a three-arm robot executing a multi-cargo transportation task in water. **b** Top view photographs showing the robot navigated to the designated location close to the cargoes and subsequently captured the three cargoes in an order; the robot then transported the cargoes to the original location within 18 min. Scale bar, 8 mm. In this scenario involving multiple robotic operations, the hydrogel robot was pre-adhered on a stainless steel disc, which served as an anchor body to avoid the undesirable adhesion between the hydrogel and the ground. **c** Schematic illustration of an adhesive robot executing a blood vessel repair task by applying an infrared (IR) light and a cylindrical permanent magnet. Created with BioRender.com. **d** Experimental demonstration of the robot navigating through the artificial blood vessel and moving to the leakage under the guidance of the magnet (left panels; scale bars, 25 mm); and of the robot elongating its body to cover the vessel damage under the application of the IR light and magnetic field (right panels). The bottom photograph of the right panels indicates a successful leakage repair without bleeding. Scale bars in the upper four photographs, 10 mm; scale bar in the bottom photograph, 25 mm. Data in (**b**) and (**d**) are representative of two independent experiments.

removing the IR light and magnet, no further blood leakage was observed for at least another week.

Although many efforts have been made to develop hydrogel robots, it is still a challenge to achieve multiple desirable performances for realistic applications[46,47]. Response time is such a key performance index that needs to be considered. In this study, robotization of the protein hydrogels involves photothermal heating, shape morphing, adhesion and cooling. Notably, thermal heating and adhesion can be completed within a short time (40 s; Fig. 4d), yet ~3–8 min may be required for the magnetic field-driven shape morphing (Figs. 5c and 6b) and cooling of the robot for strong adhesion (Fig. 5c). The slow shape transformation may be due to the relatively weak magnetic field used and the low content of $Fe_3O_4$ particles incorporated into the gel. In addition, after cooling of the elongated hydrogel robot, its shape could not completely recover to the original state. Nonetheless, due to the dynamic nature of the electrostatic interactions, the deformed robot could be thermally remolded into on-demand shapes for another task, reflecting potentially good reusability of the protein hydrogel robot.

In summary, we have designed and constructed a protein hydrogel-type adhesive for bionic underwater adhesive robots. The hydrogel was prepared through the complexation of positively charged, chemically crosslinked R32 and polyanionic SiW. With introduction of the dynamic electrostatic interactions, the hydrogel was endowed with a sharp phase-transition at room temperature. Therefore, the gel could realize reversible transformation from an adhesive soft state (at a temperature higher than $T_s$) to a non-adhesive rigid state (at a temperature lower than $T_s$) in water upon heating and cooling within a narrow and mild temperature range. More intriguingly, the hydrogel could achieve strong underwater adhesion at a relatively low temperature by warm attaching with target surfaces in water. Based on these features, an underwater adhesive robot was successfully fabricated by incorporating magnetic $Fe_3O_4$ nanoparticles into the gel that displayed both photothermal and magnetic field-driven responses. Under the cooperative control of the IR light and magnetic field, the robot has been demonstrated to be capable of capturing and delivering multiple cargoes from a distance in water, and repairing artificial blood vessel leakage. We anticipate that this work can provide a good model for the future development of bionic intelligent underwater robot.

## Methods

### Biosynthesis of R32 protein
The resilin-like protein R32 was recombinantly biosynthesized in bacterium *Escherichia coli* BL21(DE3) harboring plasmid pET19b-R32 (ref. 48). Briefly, the recombinant *E. coli* cells were grown in rich

medium by shake flask cultivation for protein expression. The target R32 protein was purified from cell lysate by using immobilized-metal-affinity chromatography, extensively dialyzed in deionized water, and freeze-dried before storage or use[48]. The purity of the protein was analyzed via 10% sodium dodecyl sulfate-polyacrylamide gel electrophoresis (SDS-PAGE) with Coomassie staining, and its molecular weight was confirmed by matrix-assisted laser desorption ionization time-of-flight (MALDI-TOF) mass spectrometry (Autoflex Speed; Bruker Daltonics, Leipzig, Germany). The complete amino acid sequence of this resilin-like protein is shown as Supplementary Note 1.

## Hydrogel preparation

Two types of protein hydrogels were prepared. In one experimental setup, lyophilized R32 protein was first dissolved in deionized water with mixing. Following addition of stock solutions of HRP (enzyme activity of 15,000 U ml$^{-1}$) and hydrogen peroxide (3% in H$_2$O), the resulting mixture was mildly shaken to form a uniform solution, which contained the protein, HRP, and H$_2$O$_2$ at a final concentration of 200 mg ml$^{-1}$, 600 U ml$^{-1}$ and 0.03%, respectively. This reaction mixture was then transferred into a 10-ml plastic syringe with a blunt metal needle of 0.5 mm in diameter and incubated at 37 °C for varying time periods (0–20 min) to yield di-tyrosine crosslinked protein solution. This solution was then pumped at room temperature and a flow rate of 0.5 ml min$^{-1}$ into a 20-ml aqueous bath containing SiW at 50 mg ml$^{-1}$. The tip of the syringe needle was placed 30 cm above the SiW bath surface, and the generated liquid droplets each had a volume of ~20 μl. Notably, due to the fast complexion of R32 protein with SiW, the droplets quickly transformed into spherical complexes which were scattered to float on the surface of the SiW bath. Upon stirring for 30 s, the individual complexes turned into a fibrous mat, and stirring for additional 10 min allowed the formation of a bulk hydrogel around the stirring bar. This hydrogel was then taken out and blotted with filter paper to remove surface moisture before extensive characterizations.

In another experiment, photothermally and magnetically responsive hydrogels were prepared in a similar manner except that magnetic Fe$_3$O$_4$ nanoparticles (Catalog No. M814132; Macklin Inc., Shanghai, China) were incorporated into the gels. Briefly, the monodispersed Fe$_3$O$_4$ nanoparticles were added, with stirring, into the R32 protein solutions with desirable degrees of prior di-tyrosine crosslinking. The resulting colloidal suspension with nanoparticle concentration of 15.5 mg ml$^{-1}$ was pumped into the SiW bath for gel formation. In this scenario, the Fe$_3$O$_4$ nanoparticles were fully incorporated into the ultimate hydrogels, which contained the nanoparticles at ~6% (w/w).

## Hydrogel characterization

The water content of the hydrogel was determined by measuring the weights of the hydrogel specimens before and after lyophilization. The water content was then calculated using the following equation:

$$Water\ content(\%) = \frac{W_i - W_d}{W_i} \times 100 \qquad (1)$$

where $W_i$ is the initial weight of the hydrogel before freeze drying and $W_d$ is the weight of the lyophilized hydrogel.

Stability and degradation of the hydrogels in water were studied by monitoring the weight changes of the gels during soaking in deionized water over a wide range of temperatures (10–50 °C).

Thermogravimetric analysis (TGA) of the lyophilized hydrogels was performed on an SDT Q600 thermal analyzer (TA Instruments, New Castle, DE, USA) in flowing nitrogen from 25 to 900 °C with a heating rate of 10 °C min$^{-1}$. The mass loss from 25 to 200 °C was attributed to the loss of crystal water in the lyophilized hydrogel, the mass loss from 200 to 900 °C was attributed to the decomposition of the protein R32, while the residual mass at 900 °C was attributed to

the inorganic species (SiO$_2$ and WO$_3$)[26]. The mass percent composition was then used to estimate the molar ratio of SiW to R32 in the hydrogels.

The di-tyrosine crosslinking degree of the hydrogels was measured by using a fluorescence-based method[49]. Briefly, the hydrogels were lyophilized and then hydrolyzed in 6 M HCl supplemented with 0.5% (v/v) phenol under nitrogen gas protection. The hydrolysates were blow-dried with nitrogen and re-dissolved in 100 mM phosphate buffer (pH 7.2) for fluorescence measurements (excitation 315 nm, emission 410 nm). An authentic di-tyrosine standard was dissolved and diluted in the same phosphate buffer for fluorescence measurements. The di-tyrosine crosslinking degree was presented as a percentage of the determined level of di-tyrosine in each hydrogel hydrolysate to the stoichiometric amount in the hydrogel.

Scanning electron microscopy (SEM) characterization of the lyophilized hydrogels was performed on a Hitachi S-3400N scanning electron microscope (Hitachi, Tokyo, Japan). To prepare the specimens, the samples were coated with gold using a Leica EM SCD050 sputtering device with a water-cooled sputter head (Leica Microsystems GmbH).

The underwater oil contact angles of the hydrogels were measured on the DSA100 Drop Shape Analyzer (KRÜSS Scientific, Hamburg, Germany). Disc-shaped hydrogel samples with a diameter of 20 mm and a thickness of 5 mm were immersed in deionized water at ambient temperatures as specified, and no air bubbles were allowed to remain on the testing surfaces. The needle of the syringe for oil supply was immersed into the water, and then 5 μl 1,2-dichloroethane was dispensed and slightly shaken down onto the sample surface.

## Mechanical properties tests

The rheological behavior was investigated on a TA Instruments AR-G2 rheometer with a 40 mm parallel-plate geometry. For studying dynamics of di-tyrosine crosslinking of R32 protein, time-sweep tests were performed at 37 °C with a strain of 1% and a frequency of 1 Hz. To load the samples, 500 μl of the mixture with HRP (final enzyme activity of 200 U ml$^{-1}$), 0.01% w/w hydrogen peroxide, and protein at a final concentration of 200 mg ml$^{-1}$ was transferred onto the prewarmed bottom plate at 37 °C. The top plate was then lowered to a gap distance of 310 μm, and hydrogenated silicone was added around the circumference to minimize dehydration. For studying response times of the switch in hydrogel stiffness, time sweeps were performed for the R32-5%-SiW hydrogel upon temperature shifts between 20 and 30 °C, at a strain of 1% and a frequency of 1 Hz. In another experimental setup, frequency sweeps of the R32-5%-SiW hydrogel were performed at a constant strain of 0.3% and temperatures at 20 °C and 30 °C, respectively.

Temperature-dependent tensile testing of the hydrogels was performed on an Instron 5944 testing machine equipped with a 10 N load cell (Instron Corporation). For the temperature control, a water bath was used. The tensile tests were conducted on the dumbbell-shaped gels (overall length, 20 mm; width, 6 mm; inner width, 2 mm; gauge length, 10 mm; and thickness, 2 mm), and the deformation rate was set at 10 mm min$^{-1}$.

The dynamic mechanical tests of the hydrogels (length of 20 mm, width of 5 mm, and thickness 2 mm) were conducted on a Q800 dynamic mechanical analyzer (TA Instruments). For all the samples, dynamic temperature sweep measurements were performed in a submersion tensile mode at 0.1% strain, 1 Hz frequency and a preload force of 0.1 N. The temperature was ramped from 0 to 80 °C at the rate of 3 °C min$^{-1}$.

For the self-healing tests, the freshly prepared hydrogels were first molded into a dumbbell shape (overall length, 35 mm; width, 6 mm; inner width, 2 mm; gauge length, 15 mm; and thickness, 2 mm). Afterwards, the dumbbell-shaped gels were cut into two halves in the middle, and gently transferred into the original mold. These gels were

subsequently healed in a water bath at different temperatures for different durations followed by tensile testing at 25 °C.

## Underwater probe tack test

The probe tack tests, for measuring the adhesion strength of the hydrogels, were performed on an Instron 5944 testing machine with a 100 N load cell in a water bath. The water bath was equipped with a temperature controller, which allows for the measurement of the adhesiveness under different temperature conditions. Hydrogel samples were molded into a disc shape with 10 mm in diameter and 1 mm in thickness. In a typical adhesion experiment, the hydrogel sample was first adhered on a target substrate that anchored to the bottom of water bath. Then, the probe substrate fixed on the load cell was brought down to contact with the gel with a preload of 0.5 N for 30 s, which corresponded to a compressive stress of ~6 kPa. The probe substrates included stainless steel, polypropylene (PP), glass, paper, rubber, bone, heart, muscle, kidney and liver. After that, the detachment was performed and the adhesion strength was determined by adhesion force curves. Specifically, the test velocity was set to be 50 mm min$^{-1}$ in the tensile mode to obtain the load-displacement curve.

## Photothermal test

The photothermal properties of the hydrogels in water were measured via the following procedure. Briefly, disc-shaped hydrogel samples with a diameter of 6 mm and a thickness of 1 mm were made and then submerged 5 cm below the surface of a water bath. The original temperature of the water bath was set at 20 °C, and the distance between the hydrogel upper surface and an IR light bulb was set at 5 cm, 7 cm, and 9 cm, respectively. Upon illumination by the IR light lamp (600 W), the temperature of the gel specimens was monitored by a model Fotric 223 s thermal camera (Fotric system, shanghai, China).

## Photothermal and magnetic controlling experiment

Photothermal-heating by irradiation with light from the top was carried out using the IR light lamp (600 W). The light source was placed within 20 cm from the hydrogel samples. A cylindrical NdFeB magnet (4 cm in diameter and 8 cm in height) was used to apply magnetic fields required for actuation at distance. For magnetic elongation and navigation, the direction and strength of the applied magnetic fields were varied by manually manipulating the magnet to change its position and orientation.

For artificial blood vessel repairing, an S-shaped PMMA tube (inner diameter of 20 mm and wall thickness of 2 mm) was used as the model vessel and a 25-mm-long incision was produced at the bottom of the tube to mimic a wound. After filling with pig blood (37 °C), a CTI robot (length of 10 mm, width of 10 mm, and thickness of 5 mm) was inserted into the tube for wound repairing. The movement and elongation of the hydrogel robot was controlled by a permanent magnet, and the photothermal heating was triggered by the IR light irradiation.

## Statistical analysis

Information regarding error bars, number of biological replicates or samples, number of independent experiments and statistical analyses are described in the corresponding table and figures.

## Reporting summary

Further information on research design is available in the Nature Portfolio Reporting Summary linked to this article.

## Data availability

All data supporting the findings of this study are available within the article and its Supplementary files. Data are available from the corresponding authors upon request. Source data are provided with this paper.

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

## Acknowledgements

Financial support was provided by the National Key Research and Development Program of China (Grant no. 2020YFA0907702 to X.-X.X.), the National Natural Science Foundation of China (Grant nos. 22075179, 32270107 to Z.-G.Q. and grant no. 32071414 to X.-X.X.), and the Natural Science Foundation of Shanghai (21ZR1432100 to Z.-G.Q.). The authors thank Fang Pan for her assistance in preparation of the Movies.

## Author contributions

X.-X.X., Z.-G.Q. and S.-C.H. conceived the project, designed the research, analyzed the data, and wrote the manuscript. S.-C.H. performed material fabrication, characterization and demonstration experiments, analyzed the data, and wrote the draft. Y.-J.Z. and X.-Y.H. performed recombinant protein production and purification. All the authors approved the manuscript.

## Competing interests

X.-X.X., X.-Y.H., S.-C.H. and Z.-G.Q. have filed a patent application through Shanghai Jiao Tong University, and declare no other competing interests. The patent application title is "Preparation and applications of protein-based adhesive hydrogels for soft underwater robots" with Chinese Patent Application Number 202310955209.2. The specific aspects of the manuscript encompassed by the patent application include methods for fabrication and applications of the adhesive protein hydrogels. Y.-J.Z. declares no competing interests.
