## [Peer Review File · Nature Communications]

Programmable adhesion and morphing of protein hydrogels for underwater robotsREVIEWER COMMENTS

Reviewer #1 (Remarks to the Author):

This manuscript reported a protein hydrogel-type adhesive for bionic underwater adhesive robots through the complexation of intrinsically-disordered resilin-like proteins (RLPs) and Keggin-type polyoxometalates. Overall, the article is logically coherent and well written. I would recommend the publication of this paper if the following questions can be properly solved.

1. Detail missing. The ratio of RLPs and polyoxometalates affects the adhesive property of the hydrogel, what is the ratio? How is the effect of the RLPs and polyoxometalates ratio on the T_s and T_i of the hydrogel? What is the water content of the hydrogel?
2. The hydrogels "could rapidly switch between soft adhesive and stiff non-adhesive states in response to small temperature variation." What is the mechanism?
3. The introducing of Fe_3O_4 nanoparticles endow the hydrogels with photothermal and magnetic responsiveness? What is the Fe_3O_4 content inside the hydrogel? In addition, the photothermal properties of the hydrogel should be examined quantitatively.
4. Typo errors. Reference 7 and 10, the journal names are misspelled.

Reviewer #2 (Remarks to the Author):

The manuscript "Programmable adhesion and morphing of protein hydrogels for underwater robots" reports interesting protein hydrogels that are formed by two types of bonds: di-tyrosine crosslinks and complexation by polyanionic silicotungstic acid. The latter is temperature dependent, and is reversible dissociated at a temperature T_s , slightly above room temperature. This provokes a reversible and controllable fluctuation of the hydrogels between two states: stiff non-adhesive state below T_s and soft adhesive state above T_s . Differences in adhesion behavior and stiffness are remarkable in a narrow temperature window. Furthermore, when the hydrogels are doped with magnetic particles it is possible to switch them between both states upon IR illumination and the magnetic character also allows locomotion and deformation of the hydrogels by applied magnetic field. These are very original, remarkable results that make the manuscript significant in the field of polymer hydrogels and related fields, and a clear advance in relation to other papers in the field.

The manuscript is in general well-written, with appropriate references to the field and with conclusions supported by the experimental results. In relation to the methodology, the experimental techniques meet the standard in the field, and a complete characterization, specially of the mechanical properties is presented.

Because all of this, the manuscript is appropriate for publication, although some minor revision is needed before acceptance, according to the following comments:

1. The protocol for hydrogel preparation should be more detailed, so that it can be reproduced by other authors.
2. Figure S4b shows the reversibility of E' with temperature increase and decrease. The authors should

provide results for many cycles to confirm that the reversibility is maintained.

3. Since the hydrogels are intended to be used immersed in water, their stability and biodegradation in water should be studied and provided.

4. For the study of thermal healing (Figure S4), what is the compressive stress applied to allow healing? Since the mechanical properties are completely recovered under appropriate healing conditions, does this mean that the stiffness is only provided by the complexation by polyanionic silicotungstic acid? Then, what is the role of di-tyrosine crosslinks, since these are in principle not healed?

5. In Figure 3d, the label of the x-axis is not provided.

6. The microscopic mechanisms responsible of adhesion are not discussed, and this is an important thing in the context of the manuscript.

7. Does the compressive stress during adhesion influences the adhesion strength?

8. There are three main limitations of the actuation of the magnetic hydrogels. First, the response is very slow and requires time for switching from non-adhesive to adhesive, then applying the field, and finally cooling to make the hydrogel rigid again (see for example Figure 5). Secondly, after the hydrogel is elongated to capture a cargo, the initial shape cannot be recuperated (see for example Figure 5c and Figure 6b). This means that it would be possible to use the hydrogel only once in one of these tasks. Finally, as observed in Figure 6b, there is need of an anchor body for the actuation. The authors should discuss about these limitations.

Reviewer #3 (Remarks to the Author):

The authors describe a method for creating soft robots that can be controlled remotely using infrared light and a magnetic field. The robots are made from a protein (resilin-like polypeptide) hydrogel that can switch between soft and stiff states in response to temperature changes. This allows them to adhere to surfaces and perform tasks such as navigation underwater.

Although this is a well-written paper, its contribution to soft robotics and protein engineering is limited. The authors have combined several established techniques and concepts, but this reviewer is not confident that the work will appeal to the broad readership of the Nat. Commun. At this stage, a more specialized journal is a better fit.

On the technical side, I have two minor comments:

1) The paper would have been more informative if it had included more experimental detail, especially in the section on repairing model wounds. The part of the work that describes the repairs of a model wound is conceptually interesting, but the manuscript lacks sufficient experimental details to determine whether the concept can be translated into practice. For example, it is not clear what substrate is used to prepare blood vessels.

2) The authors have inaccurately used the term LLPS. While it is true that resilin-like polypeptides can undergo temperature-triggered LLPS, the temperature-dependent responses observed in this work are not caused by LLPS. Instead, they are due to the temperature dependencies of non-covalent interactions.

Responses to the reviewers' comments: Manuscript ID NCOMMS-23-27909-T

Reviewer #1 (Remarks to the Author):

This manuscript reported a protein hydrogel-type adhesive for bionic underwater adhesive robots through the complexation of intrinsically-disordered resilin-like proteins (RLPs) and Keggin-type polyoxometalates. Overall, the article is logically coherent and well written. I would recommend the publication of this paper if the following questions can be properly solved.

Response: Thank you for the positive comments.

Comment 1: Detail missing. The ratio of RLPs and polyoxometalates affects the adhesive property of the hydrogel, what is the ratio? How is the effect of the RLPs and polyoxometalates ratio on the T_s and T_i of the hydrogel? What is the water content of the hydrogel?

Response: Many thanks for the comments. We have determined the water content and molar ratio of polyoxometalate to RLP in the hydrogels (Table S1 below). These hydrogels had comparable water contents of ~20%, and variable silicotungstic acid-to-R32 ratios that depend on prior chemical crosslinking of the protein. With an increase in the di-tyrosine crosslinking degree, the silicotungstic acid-to-R32 ratio decreased from approximately 6.4:1 to 4.9:1, concomitant with a moderate increase in the softening temperature T_s and an overall marked increase in the adhesion initiating temperature T_i .

Table S1. Compositional features and thermally responsive properties of the hydrogels^a

Hydrogel type ^b	Silicotungstic acid-to-R32 molar ratio	Water content (%)	T_s (°C)	T_i (°C)
R32-SiW	6.4 ± 0.1	22.7 ± 1.2	20.4 ± 0.3	23.7 ± 0.2
R32-5%-SiW	6.0 ± 0.1	21.5 ± 1.5	21.2 ± 0.2	23.8 ± 0.2
R32-15%-SiW	5.4 ± 0.1	20.6 ± 1.7	22.5 ± 0.4	43.5 ± 0.4
R32-30%-SiW	4.9 ± 0.2	19.1 ± 0.8	26.7 ± 0.3	52.7 ± 0.5

^aData are presented as mean \pm s.d. of $n = 3$ independent samples, and are representative of two independent experiments.

^bThe hydrogels were fabricated by complexation of silicotungstic acid (SiW) with resilin-like protein R32 with prior di-tyrosine crosslinking degrees at 0, 5%, 15%, and 30%, respectively.

We have added the experimental procedure, and the above results in the revised manuscript on page 7:

These hydrogels had comparable water content of $\sim 20\%$, and variable silicotungstic acid-to-R32 ratio that depended on the prior di-tyrosine crosslinking degree of the R32 protein (Table S1). With an increase in the chemical crosslinking degree, the silicotungstic acid-to-R32 ratio decreased from approximately 6.4:1 to 4.9:1, indicating compositional tunability of the double crosslinked hydrogels.

On page 8:

The softening temperatures (T_s) of the hydrogels, which were defined as the peak temperatures of the loss factor ($\tan \delta$) curves, were found to be in the room temperature range from ~ 20.4 to 26.7°C (Figure 2a). In addition, there is a positive relationship between the T_s and the degree of di-tyrosine crosslinking in the hydrogels (Table S1).

On page 12:

Furthermore, the temperature at which the hydrogel initiated adhesion (T_i) was determined using water bath at a temperature interval of 0.5°C (Figure S8). In general, the hydrogel T_i increased markedly with an increase in the degree of di-tyrosine crosslinking.

Comment 2: The hydrogels “could rapidly switch between soft adhesive and stiff non-adhesive states in response to small temperature variation.” What is the mechanism?

Response: We appreciate the constructive comment. To explore the mechanism, we have performed a series of experiments: 1) quantification of response time of the switch by oscillatory time sweep, 2) evaluation of chain mobility by rheological frequency sweeps, and 3) evaluation of gel surface hydrophobicity by oil contact angle (OCA) analysis. As shown in the Figure below, studies on the typical R32-5%-SiW hydrogel revealed that this

gel could reversibly switch between the stiff and soft states within 9 s in response to small temperature variation. The stiff gel at 20°C has a network with low chain mobility and relatively hydrophilic surface, whereas the soft gel at 30°C had higher chain mobility and more hydrophobic surface. These significant variations in chain mobility and surface hydrophobicity contribute to the switchable adhesion behavior.

Figure S9. Switch of the hydrogel between soft adhesive and stiff non-adhesive states. (a) Time sweeps of the storage modulus E' of the R32-5%-SiW hydrogel upon switching the temperature between 20 °C and 30 °C. (b) Rheological frequency sweeps of the R32-5%-SiW hydrogel with a constant strain of 0.3% at 20 °C and 30 °C, respectively. (c) Oil contact angle analysis of the underwater R32-5%-SiW hydrogel at 20 °C and 30 °C, respectively. (d) Schematic illustration of the proposed switching mechanisms. Data in a-c are representative of two independent experiments. Data in c (right) are presented as mean \pm s.d. of $n = 3$ independent samples, with individual data points shown as purple dots.

We have added the experimental procedures, presented the new findings, and discussed the switching mechanisms on pages 13-14:

Considering both the mechanical properties and underwater adhesion, we found that the R32-5%-SiW hydrogel could switch between soft adhesive and stiff non-adhesive states.

Indeed, this gel reversibly switched between the stiff and soft states within 9 s in response to a temperature variation of 10 °C (Figure S9a). Such rapid switch in mechanical properties might be attributed to the fast dissociation-reassociation of dynamic electrostatic interactions between SiW and the positively charged residues of R32 protein. We also found that the soft gel network was more dynamic with higher chain mobility than its rigid counterpart (Figure S9b). In addition, the surface of the underwater soft gel at 30 °C was fairly hydrophobic, with an oil contact angle of only 43° (Figure S9c), whereas the gel surface was relatively hydrophilic at 20 °C.

Based on the above results, we propose mechanisms that underlie the thermally switchable adhesion behavior of the R32-5%-SiW hydrogel (Figure S9d). It is reasoned that the low chain mobility of the hydrogel at the stiff state hampers its close contact to the substrate surface, and the hydrophilic surface of the stiff hydrogel would trap a significant amount of water leading to shielding of protein adhesive groups to closely interact with the substrate surface. In contrast, the thermally-triggered switch of the gel into the soft state with high chain mobility and hydrophobic surface would benefit initiation of strong interfacial adhesion to the substrate of interest. Collectively, the significant variations in chain mobility and surface hydrophobicity contribute to the switchable adhesion behavior.

Comment 3: The introducing of Fe₃O₄ nanoparticles endow the hydrogels with photothermal and magnetic responsiveness? What is the Fe₃O₄ content inside the hydrogel? In addition, the photothermal properties of the hydrogel should be examined quantitatively.

Response: Many thanks for the constructive comments. The content of Fe₃O₄ nanoparticles was 6%, which endowed the hydrogels with photothermal and magnetic responsiveness. As suggested, we have quantified the photothermal properties of the M-R32-5%-SiW hydrogel upon IR light irradiation (see Figure S10 below). The temperature of the underwater gel could be elevated by 12 to 33°C within 3 min, reaching temperature values that well surpassed its T_s and T_i .

Figure S10. Evaluation of the photothermal properties of the M-R32-5%-SiW hydrogel. Evolution of temperature of the underwater disc-shaped gel was monitored upon IR light illumination, from varied distances between the light bulb and gel specimens as indicated in the figure. Data are presented as mean \pm s.d. of $n = 3$ independent samples, and are representative of two independent experiments.

We have added the photothermal test in the Methods, and included the above result in the main text as follows (page 13):

The resultant hydrogel was denoted as M-R32-5%-SiW, and its photothermal performance in water was quantified upon IR light illumination (Figure S10). The temperature of the underwater gel could be elevated by 12 to 33°C within 3 min, reaching temperature values that well surpassed its T_s and T_i . This feature is of particular interest as it would allow the gel to morph and adhere in a remotely controllable manner.

Comment 4: Typo errors. Reference 7 and 10, the journal names are misspelled.

Response: Thanks for pointing out the errors, and now they have been corrected (page 31).

[REVIEWER #2]

Comments: The manuscript "Programmable adhesion and morphing of protein hydrogels for underwater robots" reports interesting protein hydrogels that are formed by two types of bonds:

di-tyrosine crosslinks and complexation by polyanionic silicotungstic acid. The latter is temperature dependent, and is reversibly dissociated at a temperature T_s , slightly above room temperature. This provokes a reversible and controllable fluctuation of the hydrogels between two states: stiff non-adhesive state below T_s and soft adhesive state above T_s . Differences in adhesion behavior and stiffness are remarkable in a narrow temperature window. Furthermore, when the hydrogels are doped with magnetic particles it is possible to switch them between both states upon IR illumination and the magnetic character also allows locomotion and deformation of the hydrogels by applied magnetic field. These are very original, remarkable results that make the manuscript significant in the field of polymer hydrogels and related fields, and a clear advance in relation to other papers in the field.

The manuscript is in general well-written, with appropriate references to the field and with conclusions supported by the experimental results. In relation to the methodology, the experimental techniques meet the standard in the field, and a complete characterization, specially of the mechanical properties is presented. Because all of this, the manuscript is appropriate for publication, although some minor revision is needed before acceptance, according to the following comments

Response: Many thanks for the very positive comments.

Comment 1: The protocol for hydrogel preparation should be more detailed, so that it can be reproduced by other authors.

Response: Thank you for the constructive advice, and we have revised the hydrogel preparation procedure as follows (page 25):

Two types of protein hydrogels were prepared. In one experimental setup, lyophilized R32 protein was first dissolved in deionized water with mixing. Following addition of stock solutions of HRP (enzyme activity of 15000 U mL^{-1}) and hydrogen peroxide (3% in H_2O), the resulting mixture was mildly shaken to form a uniform solution, which contained the protein, HRP, and H_2O_2 at a final concentration of 200 mg mL^{-1} , 600 U mL^{-1} and 0.03%, respectively. This reaction mixture was then transferred into a 10-mL plastic syringe with a blunt metal

needle of 0.5 mm in diameter and incubated at 37 °C for varying time periods (0-20 min) to yield di-tyrosine crosslinked protein solution. This solution was then pumped at room temperature and a flow rate of 0.5 mL min⁻¹ into a 20-mL aqueous bath containing silicotungstic acid (SiW) at 50 mg mL⁻¹. The tip of the syringe needle was placed 30 cm above the SiW bath surface, and the generated liquid droplets each had a volume of ~20 μL. Notably, due to the fast complexation of R32 protein with SiW, the droplets quickly transformed into spherical complexes which were scattered to float on the surface of the SiW bath. Upon stirring for 30 s, the individual complexes turned into a fibrous mat, and stirring for additional 10 min allowed the formation of a bulk hydrogel around the stirring bar. This hydrogel was then taken out and blotted with filter paper to remove surface moisture before extensive characterizations. In another experiment, photothermally and magnetically responsive hydrogels were prepared in a similar manner except that magnetic Fe₃O₄ nanoparticles (Catalogue No. M814132; Macklin Inc., Shanghai China) were incorporated into the gels. Briefly, the monodispersed Fe₃O₄ nanoparticles were added, with stirring, into the R32 protein solutions with desirable degrees of prior di-tyrosine crosslinking. The resulting colloidal suspension with nanoparticle concentration of 15.5 mg mL⁻¹ was pumped into the SiW bath for gel formation. In this scenario, the Fe₃O₄ nanoparticles were fully incorporated into the ultimate hydrogels, which contained the nanoparticles at ~6% (w/w).

Comment 2: Figure S4b shows the reversibility of E' with temperature increase and decrease. The authors should provide results for many cycles to confirm that the reversibility is maintained.

Response: We appreciate the comment. Yes, the reversibility of E' with temperature increase and decrease was maintained for at least 15 cycles, as shown by the successive cyclic temperature sweeps. The updated figure is shown below.

Figure S5. Dynamic mechanical analysis (DMA) of the hydrogels. (a) The storage (E') and loss (E'') modulus of the chemically crosslinked R32 hydrogel as a function of temperature from 0 to 80 °C at a heating rate of 3 °C min⁻¹. (b) The storage (E') modulus of the R32-5%-SiW hydrogel as a function of temperature in the cyclic DMA measurements, revealing that the gel maintained the reversibility of E' for at least 15 heating-cooling cycles. Data in a and b are representative of two independent experiments.

Comment 3: Since the hydrogels are intended to be used immersed in water, their stability and biodegradation in water should be studied and provided.

Response: Thank you for the constructive comment. This has been studied by monitoring the weight change of the gels during soaking in water at different temperatures (see Fig. S4 below). We found that all the gels were very stable in water at ambient and lower temperatures. At the body and higher temperatures, these gels well retained their weights within the first 10 h, yet extended soaking eroded the gels to extents that depend on the di-tyrosine crosslinking of the gels.

Figure S4. Stability and degradation test of the hydrogels in water. The four types of hydrogels with varying degrees of di-tyrosine crosslinking were immersed in deionized water and incubated at the indicated temperatures over an extended time period of 240 h. The time courses of the gel weight change percentages are shown. Data are presented as mean \pm s.d. of $n = 3$ independent samples, and are representative of two independent experiments.

We have included the experimental procedure, and presented the new results on page 7: Furthermore, we studied stability and degradation of these hydrogels in water (Figure S4). Interestingly, all the gels were very stable in water over an extended time period of 240 h at the ambient and lower temperatures. At the body and higher temperatures, these hydrogels well retained their weights within the first 10 h, yet extended soaking partially eroded the hydrogels to extents that depended on the di-tyrosine crosslinking degrees. Overall, these results demonstrated another level of tunability (underwater stability) of the protein hydrogels by prior chemical crosslinking.

Comment 4: For the study of thermal healing (Figure S4), what is the compressive stress applied to allow healing? Since the mechanical properties are completely recovered under appropriate healing conditions, does this mean that the stiffness is only provided by the complexation by polyanionic silicotungstic acid? Then, what is the role of di-tyrosine crosslinks, since these are in principle not healed?

Response: We thank the reviewer for the comments. In the experiment for healing of the R32-5%-SiW hydrogel, no compressive stress is applied. The stiffness of this gel is deduced to be provided by the electrostatic complexation, and the contribution of di-tyrosine crosslinking to the stiffness is almost negligible.

We also agree with the reviewer that chemical crosslinking is not healable. Nonetheless, we would like to stress that di-tyrosine crosslinking of the R32 protein in solution did not result in gelation, and the purpose of this prior chemical crosslinking is to modulate the downstream complexation-triggered gelation process. As expected, many aspects of the ultimate SiW-complexed protein hydrogels, including composition (Table S1), mechanical properties (Fig. 2a,b), adhesive properties (Fig. 3a,c), thermal responsiveness (Table S1), and stability (Fig. S4) are tuned by modulating the di-tyrosine crosslinking degree.

To make it clear, we have added a note in the legend of the original Fig. S5 (now Fig. S6):

It should be noted that the pristine dumbbell-shaped hydrogels were cut into two halves, and gently transferred back into the original dumbbell mold for thermal healing without the application of external stress. Tensile testing of the pristine and healed gels were similarly performed at 25 °C.

In addition, we have revised the experimental procedure with details on page 27:

For the self-healing tests, the freshly prepared hydrogels were first molded into a dumbbell shape (overall length, 35 mm; width, 6 mm; inner width, 2 mm; gauge length, 15 mm; and thickness, 2 mm). Afterwards, the dumbbell-shaped gels were cut into two halves in the middle, and gently transferred into the original mold. These gels were subsequently healed in a water bath at different temperatures for different durations followed by tensile testing at 25°C.

Comment 5: In Figure 3d, the label of the x-axis is not provided.

Response: Thank you for the comment. The label “Dityr crosslinking degree” has been added.

Comment 6: The microscopic mechanisms responsible of adhesion are not discussed, and this is an important thing in the context of the manuscript.

Response: Many thanks for the constructive comment. We agree with the reviewer that the mechanisms that underlie switchable underwater adhesion are important to the readers. To explore the mechanisms, we have performed a series of new experiments, and the new results have been included in Fig. S9. In short, the thermally-triggered switch of the gel into the soft state with high chain mobility and hydrophobic surface would result in the following effects for activating adhesion. Firstly, the hydrogel would allow a sufficient contact with substrate surface due to compression-assisted migration and approaching of the gel network. Secondly, the hydrophobic surface of the soft gel could repel the interfacial water, thus leading to the exposure of protein adhesive groups to the counter surface and facilitating the formation of strong interfacial interaction. Finally, as the equilibrium of electrostatic interactions shifted toward the disassociated state, more protonated amine groups of the protein would be available for adhesion interaction.

As suggested, we have added the results and a paragraph of discussion on page 14.

Comment 7: Does the compressive stress during adhesion influences the adhesion strength?

Response: Yes, there is a positive link between the compressive stress and the adhesion strength, according to the additional experiment we have performed (see figure below). This might be attributed to the compression-induced migration and approaching of the adhesive groups of the protein gel to the substrate that facilitate interfacial interactions.

Figure. A positive link between the compressive stress and the adhesion strength. The R32-5%-SiW hydrogels were loaded under different compressive stresses, and the underwater adhesion strength was then determined at 25 °C, using stainless steel as the substrate.

To make it clear, we have added a note in the experimental section:

Then, the probe substrate fixed on the load cell was brought down to contact with the gel with a preload of 0.5 N for 30 s, **which corresponded to a compressive stress of approximately 6 kPa.**

In addition, we have added a note in the legend of Fig. S6, which is now displayed as Fig. S7:

The underwater adhesive strength of the R32-5%-SiW hydrogel to diverse surfaces measured by probe-tack tests. **For parallel comparison, the same compressive stress (~6 kPa) was applied in all the tests.**

Comment 8: There are three main limitations of the actuation of the magnetic hydrogels. First, the response is very slow and requires time for switching from non-adhesive to adhesive, then applying the field, and finally cooling to make the hydrogel rigid again (see for example Figure 5). Secondly, after the hydrogel is elongated to capture a cargo, the initial shape cannot be recuperated (see for example Figure 5c and Figure 6b). This means that it would be possible to use the hydrogel only once in one of these tasks. Finally, as observed in Figure 6b, there is need of an anchor body for the actuation. The authors should discuss about these limitations.

Response: Thank you for the constructive comment. We agree with the reviewer that robotization of the protein hydrogels involves photothermal heating, shape morphing, adhesion and cooling. Notably, thermal healing and adhesion can be completed within a short time (40 s; Fig. 4d); most of the time was occupied by shape morphing that required a few minutes (Fig. 5c). This may be due to the weak magnetic field used and the relatively low content of Fe₃O₄ particles incorporated into the gel. In addition, we would like to clarify that the anchor body is not indispensable, which is typically required in the scenario involving multiple robotic actuations (Fig. 6b).

To make it clear, we have added a note in the Fig. 6b legend:

In this scenario involving multiple robotic operations, the hydrogel robot was pre-adhered on a stainless steel disc, which served as an anchor body to avoid the undesirable adhesion between the hydrogel and the ground.

As suggested, we have added a short discussion about the limitations on pages 23-24:

Although many efforts have been made to develop hydrogel robots, it is still a challenge to achieve multiple desirable performances for realistic applications^{46,47}. Response time is such a key performance index that needs to be considered. In this study, robotization of the protein hydrogels involves photothermal heating, shape morphing, adhesion and cooling. Notably, thermal healing and adhesion can be completed within a short time (40 s; Fig. 4d), yet approximately 3–8 minutes may be required for the magnetic field-driven shape morphing (Fig. 5c and Fig. 6b) and cooling of the robot for strong adhesion (Fig. 5c). The slow shape transformation may be due to the relatively weak magnetic field used and the low content of Fe₃O₄ particles incorporated into the gel. In addition, after cooling of the elongated hydrogel robot, its shape could not completely recover to the original state. Nonetheless, due to the dynamic nature of the electrostatic interactions, the deformed robot could be thermally remolded into a new one with on-demand shapes for another task, reflecting potentially good reusability of the protein hydrogel robot.

[REVIEWER #3]

Comments: The authors describe a method for creating soft robots that can be controlled remotely using infrared light and a magnetic field. The robots are made from a protein (resilin-like polypeptide) hydrogel that can switch between soft and stiff states in response to temperature changes. This allows them to adhere to surfaces and perform tasks such as navigation underwater.

Although this is a well-written paper, its contribution to soft robotics and protein engineering is limited. The authors have combined several established techniques and concepts, but this reviewer is not confident that the work will appeal to the broad readership of the Nat. Commun. At this stage, a more specialized journal is a better fit.

Response: We thank the reviewer for the comments. We would like to clarify that it is novel to create soft robots with actuation mechanisms relying on dynamic underwater adhesion and morphing capability. In this study, we report the design and fabrication of such protein hydrogel robots with muscle-like stiffness at megapascal level, high extensibility, switchable underwater adhesion, and multiple responsiveness. To the best of our knowledge, this is the first report on the observation of a unique combination of desirable physicochemical properties and multiple functionalities from protein hydrogels, even though people have long expected to do so. We also find that prior chemical crosslinking of the tyrosine residues of the protein serves as a handle to tune complexation of its arginine residues with POM, and therefore the composition and properties of the ultimate protein hydrogels. In addition, we have provided insights on the mechanisms that underlie the switchable adhesion behavior, based on extensive experiments during the revision process. We believe that the significance of these findings will be appreciated by the broad readership in fields of biopolymers, material chemistry, and soft robotics.

Comment 1: On the technical side, I have two minor comments: The paper would have been more informative if it had included more experimental detail, especially in the section on repairing model wounds. The part of the work that describes the repairs of a model wound is

conceptually interesting, but the manuscript lacks sufficient experimental details to determine whether the concept can be translated into practice. For example, it is not clear what substrate is used to prepare blood vessels.

Response: Many thanks for the constructive comments. We have thoroughly revised the Method section and added more details on hydrogel preparation and characterization, self-healing, photothermal, and robotic repairing tests, according to this and the other reviewers' comments. In vessel repairing, we have actually tested several commercially available artificial blood vessels such as expanded polytetrafluoroethylene (ePTFE) blood vessel grafts and transparent polymethyl methacrylate (PMMA) tubes, and similar robotic repairing results have been observed. Considering that the artificial ePTFE vessels were opaque and difficult to visualize, we choose the transparent PMMA tubes for proof-of-concept demonstration of robotic repair within the vessels.

To make it clear, we have added the repairing test in the Methods section (page 32):

For artificial blood vessel repairing, an S-shaped polymethyl methacrylate (PMMA) tube (inner diameter of 20 mm and wall thickness of 2 mm) was used as the model vessel and a 25-mm-long incision was produced at the bottom of the tube to mimic a wound. After filling with pig blood (37°C), a CTI robot (length of 10 mm, width of 10 mm, and thickness of 5 mm) was inserted into the tube for wound repairing. The movement and elongation of the hydrogel robot was controlled by a permanent magnet, and the photothermal heating was triggered by the IR light irradiation.

In addition, we have also revised the main text in the Results section on page 21:

In order to visualize the actuation of the robot, a transparent polymethyl methacrylate (PMMA) tube was used as the artificial blood vessel. For mimicking the bleeding of the blood vessel, a 25-mm-long incision was produced at the bottom of the tube.

Comment 2: The authors have inaccurately used the term LLPS. While it is true that resilin-like polypeptides can undergo temperature-triggered LLPS, the temperature-dependent responses observed in this work are not caused by LLPS. Instead, they are due to the temperature dependencies of non-covalent interactions.

Response: We agree with the reviewer that the temperature-dependent responses of the protein hydrogels are due to the electrostatic interactions rather than LLPS. To prevent ambiguity, we choose not to introduce the LLPS feature of the protein and revised the main text as follows:

Resilin-like proteins (RLPs) are an important type of intrinsically disordered proteins which are positively charged, and chemically cross-linkable **due to the presence of characteristic tyrosine residues**^{36,37}.

Thank you and the other two referees again for all the comments.

REVIEWERS' COMMENTS

Reviewer #1 (Remarks to the Author):

The authors almost answered all the questions I am concerned. This manuscript could be published in this journal.

Reviewer #2 (Remarks to the Author):

The authors have adequately addressed my comments and those of the other reviewers. Therefore I recommend acceptance of the manuscript.